# Construction Risk Evaluation of Poor Geological Channels Based on Cloud Model-Improved AHP–Matter–Element Theory

**Qingfu Li [1], Zhipeng Wang [1,\*], Linfang Lu [2] and Qiang Ma [2]**

[1]  School of Water Conservancy Engineering, Zhengzhou University, Zhengzhou 450001, China; lqflch@zzu.edu.cn

[2]  Construction Administration of the Second Phase Project of Zhaokou Irrigation District of the Yellow River in Henan Province, Kaifeng 475000, China; zkeqllf@163.com (L.L.); maqiangluhun@126.com (Q.M.)

\*  Correspondence: wangzhipeng@gs.zzu.edu.cn

**Abstract:** In the process of economic development, the exploitation and utilization of resources has played an important role, but the subsequent post-mining collapse and the shortage of land resources have affected future reconstruction to a certain extent. Currently, there is a firm belief in sustainable development and its goals to be achieved in the future. Based on the concept of sustainable development, this paper examines the feasibility of rebuilding channels under adverse geological conditions, and studies whether there are risks and the degree of risk. According to the characteristics of the experts' judgment language and the ambiguity and randomness between various factors, it is proposed that a cloud model is used to improve the AHP (Analytic Hierarchy Process) risk assessment method. At the same time, the traditional matter–element theory is improved through the cloud model, so that the impact of uncertainty and randomness can be comprehensively considered in the evaluation; finally, forming the risk assessment system of the cloud-based AHP and cloud-based matter–elements. The application of examples shows that, compared with the methods in the relevant literature, the evaluation results of this article are more objective, more accurate, have better applicability, and play an important guiding role in channel construction under adverse geological conditions.

**Keywords:** cloud-based AHP; cloud-based matter–elements; channel construction; goaf area; sand mining area; sustainability; risk assessment

## 1. Preface

In the course of China's development, the use of resources in processes such as river sand mining and coal mining has greatly promoted economic development. However, excessive and disorderly mining, as well as the abandonment of the original land after resource mining, has also made China's land resources increasingly scarce, resulting in the subsidence of the land surface; this has caused a series of social, economic and environmental problems, which are highly unfavorable to sustainable development. With the continuous adjustment of the urban structure and the continuous strengthening, as well as the improvement, of infrastructure facilities, the demand for land available for construction is increasing day by day. Thus, the governance of old empty areas and the use of upper land are problems that need to be faced at present.

As a major agricultural country in the world, China has a long cultural history of irrigation. Since the beginning of human civilization, the way people use water resources to irrigate land has changed; with the changes and development of society, people's demand for food is increasing, alongside the associated requirements for irrigation levels. Farmland irrigation channels have been developed, with channels acting as water delivery projects for water conservancy construction. These are used to divert water from rivers, lakes, reservoirs and other water sources for agricultural irrigation, power generation, industrial and civil use, and are the most commonly used water conservancy projects. With the

continuous development of the social economy, people have proposed higher requirements for the production of crops, which has promoted the upgrading of irrigation channels. At the present stage, the scale of channel construction in China is huge, and it occupies a wide area. There are many factors that need to be considered; among them, in the Huang-Huai Plain of east Henan, China, the regional geological characteristics include mostly chalky soil and old sand mining pits left by previous regional sand mining. In East Henan, due to the long years of coal mining in the past, there are also some empty areas in the project passing area. Therefore, to discover whether the plan is feasible, and if the construction is safe, considering the many risk factors involved in the construction of the channel, it is necessary to evaluate the risk to guide construction and realize a sustainable concept at this preliminary stage.

The existence of mined-out areas and the sand-mining area damages the underground structure and causes surface deformation; thus, constructing buildings on them is very risky, and scientific and objective assessments of their stability are particularly important. At present, many scholars have conducted in-depth research on this: Zarei, E [1] used fuzzy Bayesian networks to determine the instability of the security domain system; Peng Xin [2] et al. used ANSYS to evaluate the stability of mining near large goaf areas; Kasap [3] et al. used the analytic hierarchy process to carry out a stability assessment of the open-pit coal mine; Domínguez [4] applied the decision matrix risk assessment technology to the risk assessment of underground mining in Mexico; Wang Xinmin [5] et al. used the entropy method and matter–element analysis to establish the risk of the evaluation model of the hazards relating to the mining area; Ghasemi, E. [6] et al. used a Monte Carlo simulation to quantify the safety of the goaf; Wang Wei [7] et al. used the matter–element extension method to evaluate the stability of the goaf; Guo Qingbiao [8] and others evaluated the stability of the construction site in the old goaf based on the cloud model and fuzzy analytic hierarchy process. Chen Xiaowei [9] used an orthogonal design and mathematical statistics to study the influence of different sand pit parameters on the stability of the embankment, and concluded that the distance between the sand pit and the foot of the embankment, as well as the depth of the sand pit itself, has the most significant impact on the safety of the embankment. At the same time, in the channel's construction, the stability of the channel slope should also be considered. Xu Baotian [10] used the gray clustering method to evaluate the stability grade of the control engineering slope; Chowdhury R [11] used another two methods based on the reliability theory to evaluate the stability of the slope; Fang Qiancheng [12] et al. established a slope stability classification evaluation method based on a game theory cloud model.

Although the above numerical and mathematical methods have achieved good evaluative results, the evaluation index factors in this article are uncertain; this includes the ambiguity, randomness and mutual incompatibility of the indicators, which leads to the above methods having certain limitations, having not fully considered the properties of the indicator factors. Since the cloud model can better resolve uncertainties, such as ambiguity and randomness, the matter–element analysis method is suitable for solving the incompatibility between index factors. Therefore, this paper combines the cloud model and matter–element theory to apply the cloud matter–element model to the stability assessment of the goaf and sand mining area. At the same time, the AHP method for a subjective weight analysis is also improved through the cloud model to better fit the qualitative and quantitative, fuzzy and random characteristics of expert opinions, thereby making the determination of weights more reasonable. Finally, in order to obtain a more scientific and reasonable evaluation result, it is verified through a case study, with the specific evaluation process shown in Figure 1.

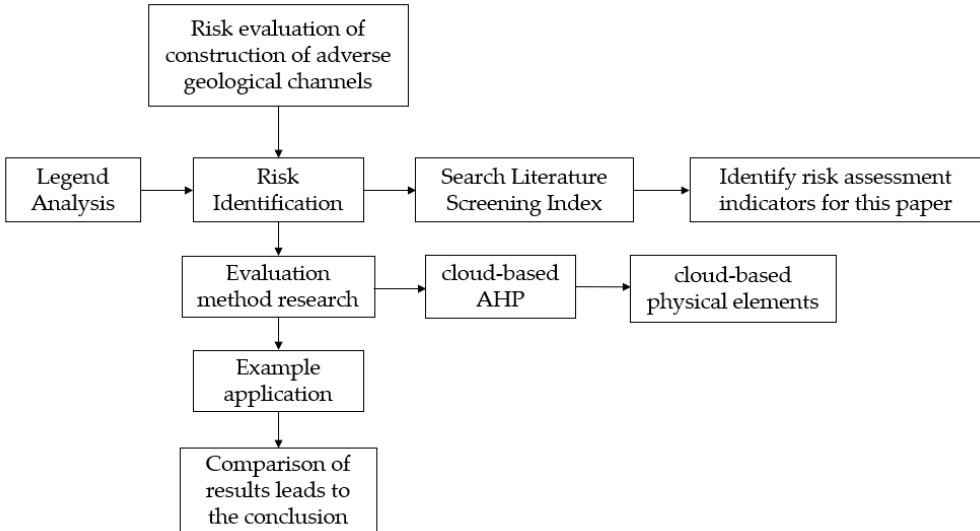

**Figure 1.** Evaluation flow chart.

## 2. Risk Identification

The first step of risk assessment was to identify risk factors. This article conducted a preliminary screening of the indicators to be evaluated by searching the relevant literature, and then extracting the risk factors that play an important role according to the research results of the literature, and determines the final indicators to be evaluated as a bad geological channel risk assessment. Based on the actual situation of this channel project, the literature on the influence of the goaf and sand mining area on new and old buildings and the risks of the building itself were studied, and preliminary evaluation indicators were obtained.

### 2.1. Risk Indicators of Goaf

The factors that affect the safety of buildings in the goaf were divided into mining factors [13], geological factors, goaf parameters, hydrological conditions and other environmental factors [14–18]. Among them, mining factors included the stop time, number of coal seams, mining coal depth, depth-to-thickness ratio and coal seam inclination. Geological factors included the rock mass structure, geological structure, rock quality indicators and joint orientation factors. Goaf parameters included the goaf buried depth, goaf span, goaf height, goaf area and the size and layout of the pillars. The hydrological conditions included groundwater factors and the degree of groundwater activity. Environmental factors included external disturbances and surrounding mining influences.

### 2.2. Risk Indicators of Sand Mining Area

The empty area left by sand mining would have a great impact on the safety of nearby river-crossing buildings. The main influencing factors were the distance between the sand pit and the building, the depth of the sand mining pit, the length of the sand mining pit and the shape of the sand pit [19–21].

### 2.3. Slope Stability Risk Indicators

The influencing factors of slope stability included the topography and geomorphology, geological environment, meteorology and hydrology and environmental factors. Among them, topography and geomorphology included the slope height, slope gradient and slope morphology [22]. The geological environment included the structural characteristics of rock mass, cohesion, angle of internal friction, fissure density and water permeability [23]. Meteorological and hydrology included the maximum rainfall in the first ten days of the year, the number of annual rainstorm days, the amount of water seepage and the

effect of the slope water flow [24]. Environmental factors included vibration and human activities [25].

According to the research results of the related literature, the factors that have an important impact were extracted, and combined with the actual situation of the project, the indicators were classified and divided. The risk factors affecting the construction of the channel were divided into the risk of goaf, external factors, risk of sand mining area and intrinsic risk of the channel. Among them, goaf risk included the stopping time of the goaf, the span of the goaf, the area of the goaf and the buried depth of the goaf; the external factors included human activities and rainfall; the risks of the sand mining area included the distance from the slope angle of the sand mining area, the shape of the sand mining pit and the depth of the sand mining pit; the inherent risks of the channel included the geological structure, groundwater, cohesion of foundation soil and slope gradient of side slopes. The resulting risk-causing hierarchy model of channel construction is shown in Figure 2.

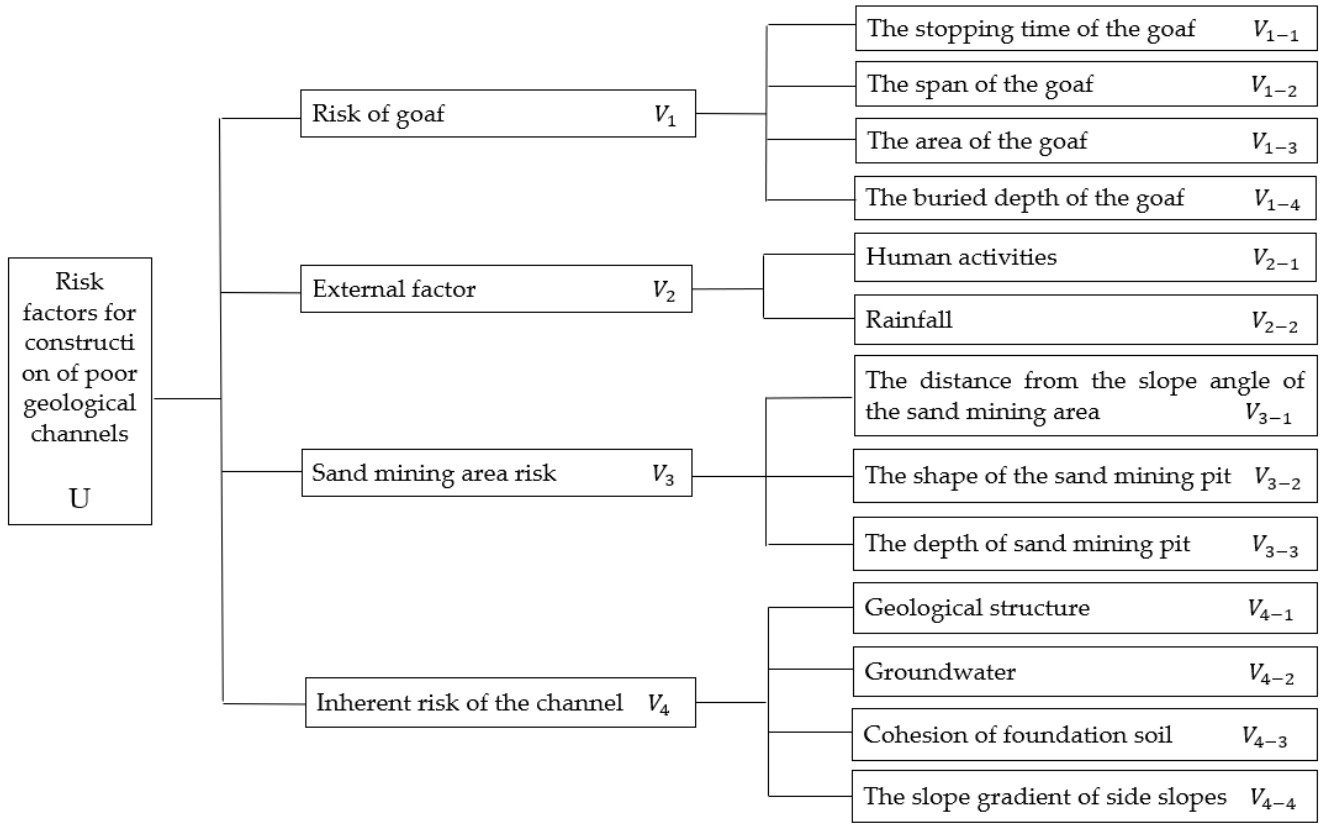

**Figure 2.** The hierarchical model of risk causes in construction of poor geological channels.

## 3. Evaluation Method

At present, the commonly used weight determination methods include the subjective weighting method, objective weighting method and subjective–objective weighting method. Among them, the subjective weighting method includes the analytic hierarchy process [4], expert survey method and feature vector method. The objective weighting method includes the fuzzy evaluation method, entropy value method [5], rough set method and weight inverse analysis method. The subjective–objective weighting method is further divided into the gray correlation degree method, compromise coefficient integrated weight method, multi-objective linear weighting function method and most unfavorable rank discriminant method. Among them, the analytic hierarchy process is widely used in various fields due to its simple operation and concise system. For example, it is used for collaborative evaluation in the company's operation management [26]; in software-defined networks,

it is used for controller selection [27]; in the design of urban tourism routes, it is used to evaluate tourism resources [28]; it is used to evaluate the importance of active power in research [29]; it is also employed to select the most suitable city when solving spatial problems [30], etc. However, the analytic hierarchy process also has its shortcomings that it cannot solve the comprehensive evaluation problem of multi-person decision-making conflicts. The constructed judgment matrix is not objective and comprehensive, and it cannot accurately reflect the subjective preference relationship of the decision maker. At the same time, it is difficult to consider the ambiguity and randomness of the problem.

Commonly used risk evaluation methods include the fuzzy comprehensive evaluation method, LEC method, Monte Carlo simulation, grey system evaluation, extension matter–element, cloud model and artificial neural network. Among them, the matter–element extension theory [23] is used to integrate the core viewpoints of the matter–element theory into the topology, using the matter–element as the basic element to describe the thing, forming an ordered triad, which is denoted as $R = (N, C, V)$, where $N$ is the name of the described thing, $C$ is the feature of the thing and $V$ is the feature value of the thing. Based on the topologic theory and matter–element theory, the topologic evaluation method can determine the classical domain and nodal domain by topologically changing the thing to be evaluated, and calculating the correlation function and correlation degree so as to carry out the evaluation process combining quantitative and qualitative, which can effectively solve the uncertainty and contradiction problems. For example, in the study of modular design methods, it is used to eliminate obstacles between theory and engineering practice [31] to achieve the comprehensive evaluation and selection of contractors in the field of engineering [32] and in the evaluation of carpool matching [33], etc. However, it is used to determine the characteristic quantity value by a specific value interval, ignoring the randomness and fuzziness of the quantity value.

In contrast, the theory of cloud model [12] describes qualitative concepts in natural language and models the transformation of uncertainty between the values given by it. It effectively integrates the randomness and ambiguity of objective things or human knowledge, and the research through unified mathematical expressions, which better reflect the universal laws of objective phenomena with randomness and ambiguity. It is used in the assessment of the development level of regional industrialized buildings [34], the assessment of the temporal and spatial variability of lake water quality [35] and the portfolio selection of variable risk preferences [36].

According to the description in the preface, it can be found that most of the current research methods are the independent use of the above three methods. This article aims to use the advantages of the cloud model to improve the traditional analytic hierarchy process and matter–element theory, in order to obtain more realistic evaluation results. Therefore, in this paper, the cloud model was used to improve the traditional analytic hierarchy process and the extension matter–element theory. The comparison scale assignment of the two risk elements in the judgment matrix can reflect the randomness, and the aggregation algorithm of the cloud model can make the decision of multiple people as all assignments are brought into the calculation formula to facilitate decision making. At the same time, the extension matter–element improved by the cloud model uses uncertainty and random reasoning mechanisms to make the evaluation results more accurate when calculating the correlation degree; the specific steps were as follows. The parameters and meanings in the formula used below were summarized in the order in which they appear, as shown in Table 1.

**Table 1.** Parameters and meanings.

| Notation | Explanation |
| --- | --- |
| $A_i$ | Cloud model structure |
| $E_{x_i}$ | Expectation |
| $E_{n_i}$ | Entropy |
| $H_{e_i}$ | Hyper entropy |
| $\beta$ | The adjustable coefficient |
| $n$ | The number of indicators to be evaluated |
| $\overline{E_x}$ | Aggregate expected mean |
| $\overline{E_n}$ | Aggregate entropy mean |
| $\overline{H_e}$ | Aggregate hyper entropy mean |
| $A_{W_i}$ | Weights |
| $A_{\overline{W_i}}$ | Relative weight |
| $E'_{x_i}$ | Expected value of relative weight |
| $E'_{n_i}$ | Relative weight entropy |
| $H'_{e_i}$ | Relative weight hype entropy |
| $I$ | Consistency ratio |
| $C$ | Consistency index |
| $R$ | Ratio index |
| $R = (N, C, V)$ | Matter–element model |
| $C_{\min}$ | Constraint interval small value |
| $C_{\max}$ | Constraint interval large value |
| $E_n^{(1)}$ | Cloud entropy based on "$3E_n$" rule |
| $E_n^{(2)}$ | Cloud entropy based on "50% association degree" rule |
| $x$ | Measured value |
| $k$ | Correlation |
| $E'_n$ | Normal random number |
| $B$ | Evaluation vector |
| $r$ | Comprehensive evaluation score |
| $b_j$ | Judging the components of the vector |
| $f_j$ | Rating value of grade $j$ |
| $j$ | Rating |
| $E_{rx}$ | Expected value of the composite judgment score |
| $E_{rn}$ | Entropy of the composite judgment score |
| $r_h(x)$ | The fraction resulting from the $h$-th operation |
| $m$ | Number of operations |
| $h$ | Number of current operations |
| $\theta$ | Reliability factor |

### 3.1. Cloud Model Improvement AHP

We used the cloud model to transform the judgment matrix in AHP. Through the 1–9 scaling method, a numerical judgment matrix was established to quantitatively represent the decisions. Assuming the existence of the theoretical domain, $U = \{\ i\ \}, i = 1, 2, 3 \ldots 9$, denoted by $A_i$, the 9-cloud models with the structure $A_i = (E_{x_i}, E_{n_i}, H_{e_i})$, $E_{x_i}, E_{n_i}, H_{e_i}$ are expectation, entropy and hyper entropy, respectively [37]. In Reference [38], the golden section method was used to calculate the above $A_i$ parameters, as shown in Equation (1):

$$
\begin{cases}
E_{x_i} = 1, 2, 3 \ldots 9, i = 1, 2, 3 \ldots 9, E_{x_i} \in Z \\
E_{n_1} = E_{n_3} = E_{n_5} = E_{n_7} = E_{n_9} = 0.382\alpha(x_{\max} - x_{\min})/6 = 0.437 \\
E_{n_2} = E_{n_1}/0.618, E_{n_4} = E_{n_3}/0.618, E_{n_6} = E_{n_5}/0.618, E_{n_8} = E_{n_7}/0.618 \\
H_{e_1} = H_{e_3} = H_{e_5} = H_{e_7} = H_{e_9} = 0.382\alpha(x_{\max} - x_{\min})/36 = 0.073 \\
H_{e_2} = H_{e_1}/0.618, H_{e_4} = H_{e_3}/0.618, H_{e_6} = H_{e_5}/0.618, H_{e_8} = H_{e_7}/0.618
\end{cases}
\tag{1}
$$

where $x_{\max} = 9$, $x_{\min} = 1$ and $\alpha$ are the adjustment factors, generally taking 0.858 [38].

From Equation (1), a 9-cloud model could be obtained, and then we used the preference aggregation of floating clouds [38] to compare the importance of the pair. The specific ideas were as follows:

Suppose there are two adjacent base clouds, $A_1$ and $A_2$, in the universe $U$, and a floating cloud $A$ representing the blank language value of its qualitative concept gap is generated between them. When $A$ moves from one cloud to the other, it is gradually reduced under the influence of the former, while the group under the influence of the latter gradually increases. Then, the parameters of floating cloud $A$ can be obtained in Equation (2):

$$
\begin{aligned}
E_x &= \beta_1 E_{x_1} + \beta_2 E_{x_2} \\
E_n &= \frac{E_{n_1}(E_{x_2} - E_x) + E_{n_2}(E_x - E_{x_1})}{E_{x_2} - E_{x_1}} \\
H_e &= \frac{H_{e_1}(E_{x_2} - E_x) + H_{e_2}(E_x - E_{x_1})}{E_{x_2} - E_{x_1}}
\end{aligned}
\tag{2}
$$

where $\beta$ is the adjustable coefficient, which is determined by the expert according to the situation. Let $\beta_1 = k_1/(k_1 + k_2)$, $\beta_2 = k_2/(k_1 + k_2)$, $\beta_1 + \beta_2 = 1$ and $k_i$ be the number of times the $i$-th cloud model is assembled. When the expert thinks that there is no need to intervene in the assembly activity, $\beta_1 = \beta_2 = 0.5$. The final numerical characteristics of the importance scaled cloud model are shown in Table 2. It can be seen from the table that the larger the expected $E_{x_i}$ value is, the more important it is.

**Table 2.** Importance scale.

| Importance Scale | Definition |
|---|---|
| $A_1 = (E_{x_1}, E_{n_1}, H_{e_1}) = (1, 0.437, 0.073)$ | $u_i$ and $u_j$ are equally important compared to each other |
| $A_3 = (E_{x_3}, E_{n_3}, H_{e_3}) = (3, 0.437, 0.073)$ | $u_i$ is slightly more important than $u_j$ |
| $A_5 = (E_{x_5}, E_{n_5}, H_{e_5}) = (5, 0.437, 0.073)$ | $u_i$ is more important compared to $u_j$ |
| $A_7 = (E_{x_7}, E_{n_7}, H_{e_7}) = (7, 0.437, 0.073)$ | $u_i$ is very important compared to $u_j$ |
| $A_9 = (E_{x_9}, E_{n_9}, H_{e_9}) = (9, 0.437, 0.073)$ | $u_i$ is extremely important compared to $u_j$ |
| $A_2 = (E_{x_2}, E_{n_2}, H_{e_2}) = (2, 0.707, 0.118)$ | |
| $A_4 = (E_{x_4}, E_{n_4}, H_{e_4}) = (4, 0.707, 0.118)$ | |
| $A_6 = (E_{x_6}, E_{n_6}, H_{e_6}) = (6, 0.707, 0.118)$ | The degree of importance is between the two adjacent clouds mentioned above |
| $A_8 = (E_{x_8}, E_{n_8}, H_{e_8}) = (8, 0.707, 0.118)$ | |

Attention: in the above table, $u_i$ and $u_j$ are important elements.

Next, the scalar comparison matrix for the clouded hierarchical analysis was built in the form of Equation (3).

$$
\begin{bmatrix}
a_{11} & a_{12} & \cdots & a_{1n} \\
a_{21} & a_{22} & \cdots & a_{2n} \\
\vdots & \vdots & \ddots & \vdots \\
a_{n1} & a_{n2} & \cdots & a_{nn}
\end{bmatrix}
=
\begin{bmatrix}
A_{11} & A_{12} & \cdots & A_{1n} \\
A_{21} & A_{22} & \cdots & A_{2n} \\
\vdots & \vdots & \ddots & \vdots \\
A_{n1} & A_{n2} & \cdots & A_{nn}
\end{bmatrix}
\tag{3}
$$

As mentioned in [38], the expectation of the elements on the diagonal of the above equation was 1, and the entropy and hyper entropy were 0; $n$ is the number of indicators to be evaluated, $a_{ji} = \frac{1}{a_{ij}}$, $A_{ji} = \frac{1}{A_{ij}} = \left( \frac{1}{E_x}, \frac{E_n}{(E_x^2)}, \frac{H_e}{(E_x^2)} \right)$.

Afterwards, the evaluation results of several experts obtained were pooled, and the average value was taken. The aggregation formula is shown in Equations (4)–(6).

$$
\overline{E_x} = \frac{1}{n} \left( \sum_{i=1}^{n} E_{x_i} \right)
\tag{4}
$$

$$
\overline{E_n} = \frac{1}{n} \left( \sum_{i=1}^{n} E_{n_i}^2 \right)^{\frac{1}{2}}
\tag{5}
$$

$$
\overline{H_e} = \frac{1}{n} \left( \sum_{i=1}^{n} H_{e_i}^2 \right)^{\frac{1}{2}}
\tag{6}
$$

We normalized the matrix, introduced the multiplication operation of cloud computing [38] and used the square root method [37] to calculate the weight $A_{W_i}(E_{x_i}, E_{n_i}, H_{e_i})$ and relative weight $A_{\overline{W_i}}\left(E'_{x_i}, E'_{n_i}, H'_{e_i}\right)$ of the factor expectation, ambiguity and randomness. The calculation of each parameter is shown in Equations (7)–(9).

$$E'_{x_i} = \frac{E_{x_i}}{\sum E_{x_i}} = \frac{\left(\prod\limits_{j=1}^{n} E_{x_{ij}}\right)^{\frac{1}{n}}}{\sum\limits_{i=1}^{n} \left(\prod\limits_{j=1}^{n} E_{x_{ij}}\right)^{\frac{1}{n}}} \tag{7}$$

$$E'_{n_i} = \frac{E_{n_i}}{\sum E_{n_i}} = \frac{\left(\left(\prod\limits_{j=1}^{n} E_{x_{ij}}\right)\sqrt{\sum\limits_{j=1}^{n} \left(\frac{E_{n_{ij}}}{E_{x_{ij}}}\right)^2}\right)^{\frac{1}{n}}}{\sum\limits_{i=1}^{n} \left(\left(\prod\limits_{j=1}^{n} E_{x_{ij}}\right)\sqrt{\sum\limits_{j=1}^{n} \left(\frac{E_{n_{ij}}}{E_{x_{ij}}}\right)^2}\right)^{\frac{1}{n}}} \tag{8}$$

$$H'_{e_i} = \frac{H_{e_i}}{\sum H_{e_i}} = \frac{\left(\left(\prod\limits_{j=1}^{n} E_{x_{ij}}\right)\sqrt{\sum\limits_{j=1}^{n} \left(\frac{H_{e_{ij}}}{E_{x_{ij}}}\right)^2}\right)^{\frac{1}{n}}}{\sum\limits_{i=1}^{n} \left(\left(\prod\limits_{j=1}^{n} E_{x_{ij}}\right)\sqrt{\sum\limits_{j=1}^{n} \left(\frac{H_{e_{ij}}}{E_{x_{ij}}}\right)^2}\right)^{\frac{1}{n}}} \tag{9}$$

We used the consistency indicators $C$ and $R$ to perform the desired consistency test, which required that Equation (10) be satisfied, where $C = \frac{\lambda_{\max}-n}{n-1}$ and $R$ are the average value of the consistency index of the homogeneous random judgment matrix, in the above equation $\lambda_{\max} \approx \frac{1}{n}\sum\limits_{i=1}^{n} \left(\frac{\sum\limits_{j=1}^{n} E_{x_{ij}} A_{W_i}}{A_{W_i}}\right)$. At the same time, when calculating the parameter $\lambda_{\max}$, the unnormalized weight cloud model $A_{W_i}$ was used, and the first parameter $E_{x_i}$ was used; when the final weight was determined, the normalized weight cloud model $A_{\overline{W_i}}$ was used. The first parameter was $E'_{x_i}$.

$$I = \frac{C}{R} < 0.1 \tag{10}$$

### 3.2. Clouded Matter–Element Model

After the weights were calculated, the risk assessment of the project could be performed. The essence of the cloud material element model [39] used in this paper was to use the cloud model to redefine and construct the matter–element extension theory, and to use the general steps of extension evaluation to describe and evaluate things. The normal cloud model $(E_x, E_n, H_e)$ was used to replace the fixed-value feature value $V$ of things, so as to realize the mathematical description of the randomness and ambiguity in the evaluation process. Specifically, it can be expressed as Equation (11):

$$R = \begin{pmatrix} N & C_1 & (E_{x1}, E_{n1}, H_{e1}) \\ N & C_2 & (E_{x1}, E_{n1}, H_{e1}) \\ \vdots & \vdots & \vdots \\ N & C_n & (E_{x1}, E_{n1}, H_{e1}) \end{pmatrix} \tag{11}$$

In the formula, $E_x$ is the expectation, which represents the position of the cloud center of gravity; $E_n$ is the entropy, which represents the fuzzy degree of the qualitative concept represented by the cloud; $H_e$ is the hyper entropy, which mainly reflects the randomness of the sample, that is, the thickness of the cloud drop on the cloud image. The cloud model

uses these three numerical characteristics to reflect the quantitative characteristics of the qualitative concept as a whole.

### 3.2.1. Determination of Optimal Cloud Entropy

The clouded matter–element model treats the classification level boundaries of the assessment index as a double constraint space $(C_{\min}, C_{\max})$ through its own fuzzy and random characteristics when dividing the security risk level interval. According to the definition of cloud expectation, the central value of the constraint interval can best represent the concept of the level, the expression of cloud expectation $E_x$ is $E_x = \frac{(C_{\min} + C_{\max})}{2}$, and the grade cloud hyper entropy $H_e$ is generally taken as a constant, and can be combined with the actual experience and uncertainty of the evaluation index. The smaller the value, the smaller the randomness of the cloud droplet membership degree and the more comparable results, but many points at the boundary would be missed. The larger the value, the greater the cloud droplet dispersion and the greater the randomness of the membership degree; it can contain more points, but the comparability would be poor, and the separation between the subordinate clouds of each level would be less obvious. In this paper, it was set as 0.08 [40].

Cloud entropy $E_n$, as a measure of the ambiguity of the state level concept, reflects the acceptable range of values and affects the accuracy of level determination. There are currently two calculation methods for determining cloud entropy, the calculation formulas are $E_n^{(1)} = \frac{C_{\max} - C_{\min}}{6}$ based on the "$3E_n$" rule and $E_n^{(2)} = \frac{C_{\max} - C_{\min}}{2.3548}$ based on the "50% association degree" rule. The former obtains adjacent grades of extension clouds that are clearly separated at the boundary, which reflects the clarity of the grade division. The adjacent grade extension cloud obtained by the latter is blurred at the boundary, indicating that the critical value belongs to the upper and lower grades at the same time, reflecting the ambiguity of the grade division. In the evaluation of indicators, the former was used for the classification of levels with clear boundaries, such as danger and safety, and the latter was used for the classification of the various levels of danger below safety [40].

### 3.2.2. Correlation of Clouded Matter–Element Model

Due to the improvement of the ambiguity and randomness when determining the grade boundary, the calculation of the correlation degree of the extension matter–element model combined with the cloud model also changed. There were three forms of relevance calculations as follows [41]:

1.  The degree of correlation between the value and the cloud matter–element

If the evaluation index is a certain value $x$, it can be regarded as a cloud drop. Bringing it into a normal cloud generator, the correlation between the value $x$ and the normal cloud model $k$ can be calculated as Equation (12):

$$k = \exp\left(-\frac{(x - E_x)^2}{2(E_n')^2}\right) \tag{12}$$

where $x$ is the value of the safety evaluation index; $E_x, E_n, H_e$ is the mathematical characteristic value of the index corresponding to the extension cloud; $E_n'$ is a normal random number with an expected value of $E_n$ and a standard deviation of $H_e$.

2.  The degree of correlation between the normal cloud and the cloud matter–elements

If the evaluation index data are the characteristics of things expressed by the normal cloud model, according to the normal cloud's cloud drop spatial distribution rules, 99.74% of the cloud droplets fall on $(E_x - 3E_n', E_x + 3E_n')$, which can be regarded as a set. Then, the

relationship between the two clouds in the formula for calculating the degree is (13), and the usability of this method can be found in the literature [41], which is not repeated here.

$$k = \frac{\left\{\left(E_x^{(1)} - 3E_n^{(1)}, E_x^{(1)} + 3E_n^{(1)}\right)\right\} \cap \left\{\left(E_x^{(2)} - 3E_n^{(2)}, E_x^{(2)} + 3E_n^{(2)}\right)\right\}}{\left\{\left(E_x^{(1)} - 3E_n^{(1)}, E_x^{(1)} + 3E_n^{(1)}\right)\right\} \cup \left\{\left(E_x^{(2)} - 3E_n^{(2)}, E_x^{(2)} + 3E_n^{(2)}\right)\right\}} \tag{13}$$

3.　The degree of correlation between interval values and cloud matter–elements

If the evaluation index data are the feature of things expressed by the interval number $(C_{min}, C_{max})$, then the interval number can be transformed into a normal cloud model, and then calculated using Equation (13).

### 3.2.3. Determine the Security Level

Since the evaluation index in this paper was a definite value, the correlation degree between the various index components of the sample to be evaluated and the standard normal cloud were calculated according to Formula (12), and a comprehensive evaluation matrix $K$ was obtained, which was in the form of Equation (14):

$$K = \begin{bmatrix} k_{11} & k_{12} & k_{13} & k_{14} & k_{15} \\ k_{21} & k_{22} & k_{23} & k_{24} & k_{25} \\ \vdots & \vdots & \vdots & \vdots & \vdots \\ k_{n1} & k_{n2} & k_{n3} & k_{n4} & k_{n5} \end{bmatrix} \tag{14}$$

where $k_{jl}$ is the cloud correlation degree between the evaluation index and the security risk level; $i = 1, 2 \ldots n$ is the number of the evaluation index. In this paper, $n$ was taken as 13, $j$ was the security risk level number, and this paper was an integer of 1–5.

Combining the weight coefficient $A_{\overline{W_i}}$ to obtain the judgment vector $B$, the calculation formula is shown in Formula (15):

$$B = A_{\overline{W_i}} \cdot K \tag{15}$$

where the evaluation index weight vector $A_{\overline{W_i}}$ is composed of the weight of each evaluation index, and the first parameter $E'_{x_i}$ was still used here; $K$ is the cloud correlation matrix between the index to be evaluated and the normal cloud of the security risk level standard.

The weighted average method was used to derive the composite judging score $r$ [38] as Equation (16):

$$r = \frac{\sum\limits_{i=1}^{5} b_i f_i}{\sum\limits_{i=1}^{5} b_i} \tag{16}$$

where $b_i$ is the component corresponding to the vector $B$; $f_i$ is the score value of the level $i$ and the scores corresponding to the judging levels I to V were taken as 5, 4, 3, 2 and 1 [38].

When calculating the degree of relevance, the existence of normal random numbers would also make the result of the calculation not unique, so it was necessary to calculate the expected value $E_{rx}$ and entropy $E_{rn}$ of the comprehensive evaluation score after multiple operations [30]. The calculation formula is as shown in Equations (17) and (18):

$$E_{rx} = \frac{r_1(x) + r_2(x) + \cdots + r_m(x)}{m} \tag{17}$$

$$E_{rn} = \sqrt{\frac{1}{m} \sum_{h=1}^{m} \left(r_h(x) - E_{rx}\right)^2} \tag{18}$$

where $m$ is the number of operations, which was taken as 100 in this article; $r_h(x)$ is the comprehensive evaluation score obtained by the $h$-th operation. The final expected value is the evaluation score that best represents the risk level of channel construction. The closer

the score is to this value, the higher the security risk level. Entropy is a measure of the dispersion of the evaluation results; the larger the value is, the more scattered the evaluation results are. At the same time, the credibility factor $\theta$ [38] is defined as in Equation (19):

$$\theta = \frac{E_{rn}}{E_{rx}} \tag{19}$$

The larger the value of $\theta$, the greater the dispersion of the evaluation results, the smaller the credibility, and, in contrast, the greater the credibility of the evaluation results. The application process of the specific evaluation method is shown in Figure 3.

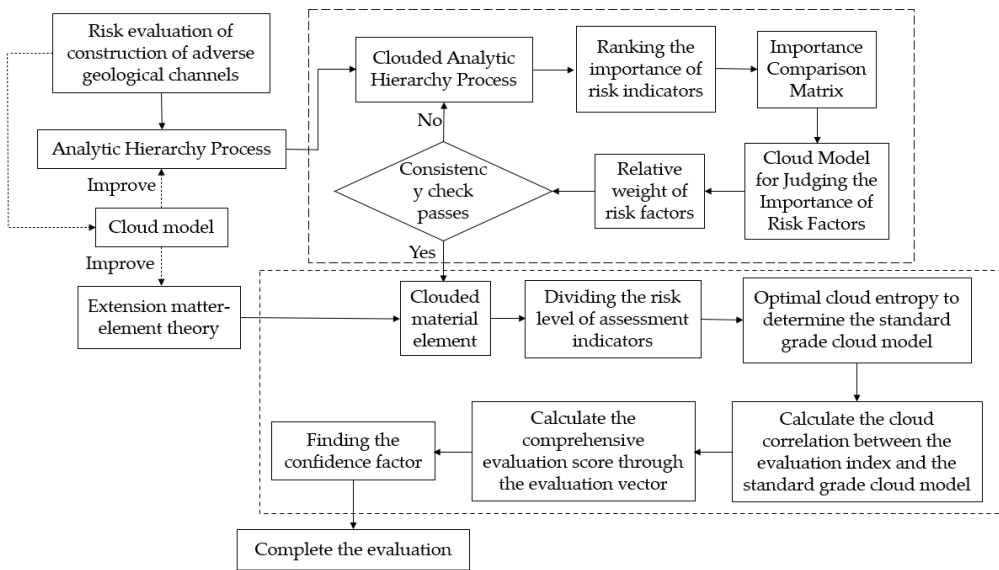

**Figure 3.** Flow chart of evaluation method.

## 4. Cloud-Based AHP and Cloud-Based Elements in the Example

### 4.1. Engineering Background

The second phase of the Zhao-Kou Yellow River Diversion Irrigation District Project in Henan Province passed through Kaifeng Wei-Shi County, Tong-xu County, Qi County and Shang-Qiu Sui County. The distribution is shown in Figure 4. This article mainly studied the Yue-Jin main canal of construction two in Qi-Xian County, mainly including the total main canal channel length of 14.4741 km (design stake number 17 + 745~32 + 219.1), the total length of channel lining of 14.3421 km and the newly built embankment road of approximately 28.95 km. The channel in this bid section was a trapezoidal section, the stake number 17 + 745~27 + 613.9 channel was designed as a single trapezoidal section, the bottom width was 23 m, the slope was 1:2.5 and the length of the lining concrete canal slope was 7.27 m. The stake number 27 + 613.9~27 + 702.6 channel was designed as a single trapezoidal section. This section was a transition section with a bottom width of 23 m~15.5 m, a side slope of 1:2 and a lining concrete canal slope length of 7.27 m~8.1 m. The stake number 27 + 702.6~32 + 219.1 channel was designed as a single trapezoidal section, with a bottom width of 15.5 m, a side slope of 1:2.5 and a lined concrete canal slope with a length of 8.1m. The instability of the local monsoon climate and the variability of the weather system caused large inter-annual precipitation differences in irrigation areas, a large disparity between the maximum and minimum precipitation and frequent inter-annual changes in high and low rainfall. The difference between the maximum and minimum precipitation in most areas was 600~1200 mm. Rainfall is distributed throughout the year in the flood season (June to September) in the irrigation area with concentrated precipitation. The average annual rainfall during the flood season was 483.8 mm. The precipitation in the four seasons is uneven, with the highest precipitation in summer from

June to August—approximately 408.9 mm; the months of the year vary greatly, the smallest precipitation month occurs in January, and the average precipitation was 11.7 mm. The largest month generally occurs in July, with an average precipitation of 186.9 mm. The channel passes through part of the goaf and sand mining area. The geology of the area is mostly silt, sandy soil, etc. The groundwater type is Quaternary loose layer pore water. The groundwater depth measured during the survey was generally 1.77~5.30 m, the water level elevation was 69.8~79.6 m. The groundwater in the project area is non-corrosive to the concrete. Sandy loam and silty sand generally have a weak to moderate water permeability; silty loam generally has a weak to weak water permeability; silty loam generally has a moderate water permeability and locally strong water permeability. The distribution of channels and the influence range of the sand mining area in the goaf are shown in Figure 5.

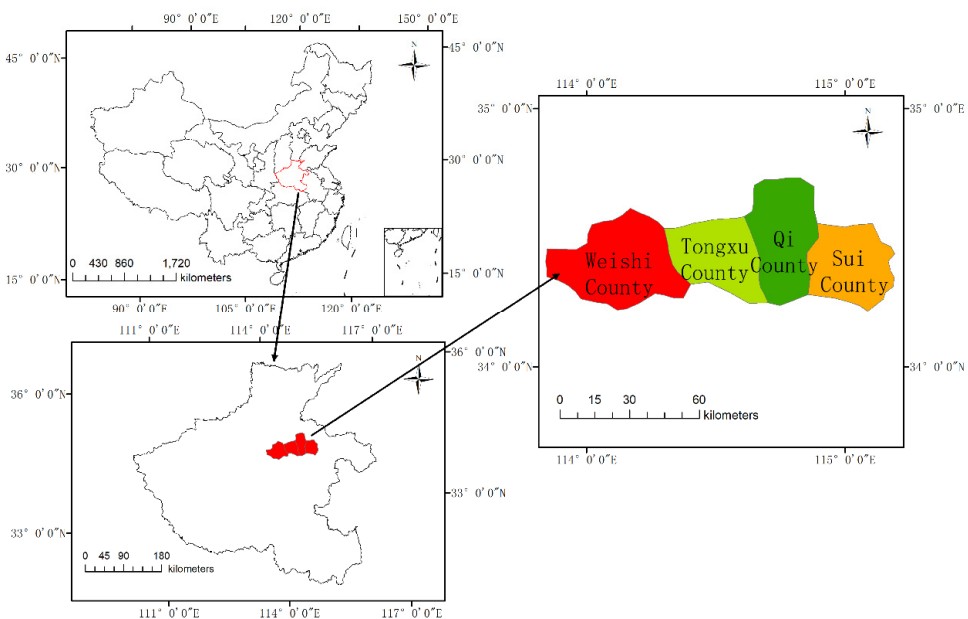

**Figure 4.** Distribution of the Yellow River irrigation district.

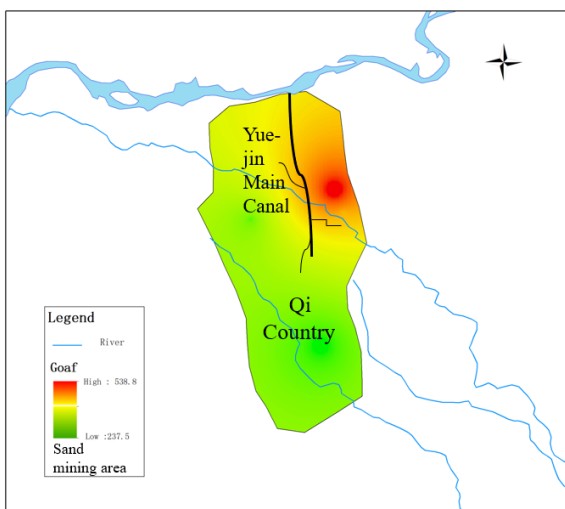

**Figure 5.** Distribution of channels and scope of influence of empty areas.

### 4.2. Determination of the Weights of Evaluation Indicators

According to project-related materials and consulting-related experts, the importance of risk indicators affecting channel construction was ranked. This article consulted three experts to compare and analyze the risk-causing factors of channel construction in turn.

The flowchart is shown in Figure 6, and the importance ranking of the first-level indicators was obtained as follows: expert one believed that the relative importance of the four risk factors was ranked as the risk of goaf $V_1$ > sand mining area risk $V_3$ > inherent risk of the channel $V_4$ > external factors $V_2$ and the risk factor comparison matrix was $Q_1$; expert two believed the relative importance of the four risk factors was ranked as the risk of goaf $V_1$ > sand-mining area risk $V_3$ = internal risk of the channel $V_4$ > external factors $V_2$ and the risk factor comparison matrix was $Q_2$; expert three believed that the relative importance of the four risk factors was the risk of goaf $V_1$ > sand mining area risk $V_3$ > inherent risk of the channel $V_4$ > external factors $V_2$ and the risk factor comparison matrix was $Q_3$; the summary results are shown in Table 3.

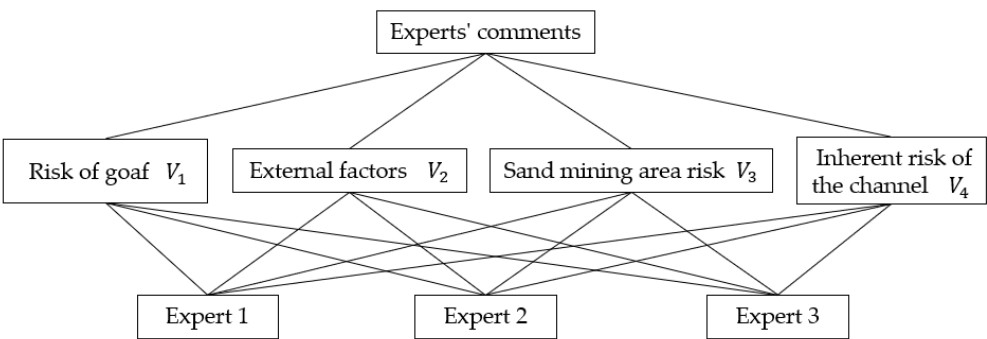

**Figure 6.** Flow chart of risk evaluation.

**Table 3.** Comparative matrix of relative importance of risk factors for the first-level indicators of adverse geological channels.

| | $Q_1$ | | | | $Q_2$ | | | | $Q_3$ | | | |
|---|---|---|---|---|---|---|---|---|---|---|---|---|
| | $V_1$ | $V_2$ | $V_3$ | $V_4$ | $V_1$ | $V_2$ | $V_3$ | $V_4$ | $V_1$ | $V_2$ | $V_3$ | $V_4$ |
| $V_1$ | 1 | 7 | 3 | 4 | 1 | 5 | 4 | 4 | 1 | 7 | 3 | 5 |
| $V_2$ | 1/7 | 1 | 1/5 | 1/4 | 1/5 | 1 | 1/4 | 1/4 | 1/7 | 1 | 1/5 | 1/4 |
| $V_3$ | 1/3 | 5 | 1 | 2 | 1/4 | 4 | 1 | 1 | 1/3 | 5 | 1 | 3 |
| $V_4$ | 1/4 | 4 | 1/2 | 1 | 1/4 | 4 | 1 | 1 | 1/5 | 4 | 1/3 | 1 |

The language judgment scales $A_{12}$ based on the cloud model given by three experts on the importance between $V_1$ and $V_2$ were $Q_{1(A_{12})} = (7, 0.437, 0.073)$, $Q_{2(A_{12})} = (5, 0.437, 0.073)$ and $Q_{3(A_{12})} = (7, 0.437, 0.073)$; the importance judgment cloud model of $V_1$ and $V_2$ obtained after the aggregation of Formulas (4)–(6) was expressed as $(6.333, 0.252, 0.042)$. The cloud models of importance judgments of other risk factors obtained by the same method are summarized in Table 4.

**Table 4.** The importance of risk factor cloud model matrix.

| $i$ | $j$ | | | |
|---|---|---|---|---|
| | 1 | 2 | 3 | 4 |
| 1 | (1,0,0) | (6.333,0.252,0.042) | (3.333,0.313,0.052) | (4.333,0.364,0.061) |
| 2 | (0.162,0.007,0.001) | (1,0,0) | (0.217,0.017,0.003) | (0.250,0.025,0.004) |
| 3 | (0.305,0.027,0.004) | (4.667,0.313,0.052) | (1,0,0) | (2.000,0.313,0.052) |
| 4 | (0.233,0.022,0.003) | (4.000,0.408,0.068) | (0.611,0.158,0.026) | (1,0,0) |

The relative weights calculated by Equations (7)–(9) are shown in Table 5.

**Table 5.** Relative weights of risk factors.

| $A_{W_i}$ | $A_{\overline{W_i}}$ |
| --- | --- |
| (3.093,1.857,1.191) | (0.555,0.524,0.527) |
| (0.308,0.186,0.119) | (0.055,0.053,0.053) |
| (1.299,0.860,0.549) | (0.233,0.243,0.243) |
| (0.869,0.639,0.400) | (0.156,0.180,0.177) |

The consistency test was performed by Equation (10) to obtain $I = 0.077 < 0.1$ and, thus, the first-level indicator risk of goaf $V_1$, external factors $V_2$, sand-mining area risk $V_3$ and internal risk of the channel $V_4$ had a weight vector of (0.555,0.055,0.233,0.156).

For the second-level indicators under the risk of the goaf, expert one believed that the relative importance of the four risk factors was ranked as the stop time of the goaf $V_{1-1}$ > the depth of the goaf $V_{1-4}$ > the span of the goaf $V_{1-2}$ > the area of the goaf $V_{1-3}$ and the risk factor comparison matrix was $Q_{1-1}$; expert two believed that the relative importance of the four risk factors was ranked as the stop time of the goaf $V_{1-1}$ > the span of the goaf $V_{1-2}$ > the depth of the goaf $V_{1-4}$ > the area of the goaf $V_{1-3}$ and the risk factor comparison matrix was $Q_{1-2}$; expert three believed that the relative importance of the four risk factors was ranked as the stop time of the goaf $V_{1-1}$ > the span of the goaf $V_{1-2}$ > the depth of the goaf $V_{1-4}$ > the area of the goaf $V_{1-3}$ and the risk factor comparison matrix was $Q_{1-3}$.

For the second-level indicators under external factors, expert one believed that the relative importance of the two risk factors was ranked as human activity $V_{2-1}$ > rainfall $V_{2-2}$ and the risk factor comparison matrix was $Q_{2-1}$; expert two believed that the relative importance of the two risk factors was ranked as human activity $V_{2-1}$ > rainfall $V_{2-2}$ and the risk factor comparison matrix was $Q_{2-2}$; expert three believed that the relative importance of the two risk factors was rainfall $V_{2-2}$ > human activities $V_{2-1}$ and the risk factor comparison matrix was $Q_{2-3}$;

For the second-level indicators under the risk of the sand mining area, expert one believed that the relative importance of the three risk factors was ranked as the distance from the slope angle of the sand mining area $V_{3-1}$ > the depth of the sand mining pit $V_{3-3}$ > the shape of the sand mining pit $V_{3-2}$ and the risk factor comparison matrix was $Q_{3-1}$; expert two believes that the relative importance of the three risk factors is ranked as the distance from the slope angle of the sand mining area $V_{3-1}$ > the depth of the sand mining pit $V_{3-3}$ > the shape of the sand mining pit $V_{3-2}$, and the risk factor comparison matrix is $Q_{3-2}$; expert three believed that the relative importance of the three risk factors was ranked as the distance from the slope angle of the sand mining area $V_{3-1}$ > the depth of the sand mining pit $V_{3-3}$ > the shape of the sand mining pit $V_{3-2}$ and the risk factor comparison matrix was $Q_{3-3}$;

For the second-level indicators under the inherent risk of the channel, expert one believed that the relative importance of the four risk factors was ranked as geological structure $V_{4-1}$ > cohesion $V_{4-3}$ > the slope gradient of side slopes $V_{4-4}$ > groundwater $V_{4-2}$ and the risk factor comparison matrix was $Q_{4-1}$; expert two believed that the relative importance of the four risk factors was ranked as geological structure $V_{4-1}$ > cohesion $V_{4-3}$ > the slope gradient of side slopes $V_{4-4}$ > groundwater $V_{4-2}$ and the risk factor comparison matrix was $Q_{4-2}$; expert three believed that the relative importance of the four risk factors was ranked as geological structure $V_{4-1}$ = cohesion $V_{4-3}$ > the slope gradient of side slopes $V_{4-4}$ > groundwater $V_{4-2}$ and the risk factor comparison matrix was $Q_{4-3}$. The summary comparison matrix of all levels of indicators is shown in Tables 6–9.

**Table 6.** Comparison matrix of relative importance of risk factors of secondary indicators under the risk of goaf.

| | $Q_{1\text{-}1}$ | | | | $Q_{1\text{-}2}$ | | | | $Q_{1\text{-}3}$ | | | |
|---|---|---|---|---|---|---|---|---|---|---|---|---|
| | $V_{1\text{-}1}$ | $V_{1\text{-}2}$ | $V_{1\text{-}3}$ | $V_{1\text{-}4}$ | $V_{1\text{-}1}$ | $V_{1\text{-}2}$ | $V_{1\text{-}3}$ | $V_{1\text{-}4}$ | $V_{1\text{-}1}$ | $V_{1\text{-}2}$ | $V_{1\text{-}3}$ | $V_{1\text{-}4}$ |
| $V_{1-1}$ | 1 | 4 | 6 | 4 | 1 | 3 | 7 | 5 | 1 | 2 | 6 | 5 |
| $V_{1-2}$ | 1/4 | 1 | 3 | 1/2 | 1/3 | 1 | 4 | 3 | 1/2 | 1 | 4 | 3 |
| $V_{1-3}$ | 1/6 | 1/3 | 1 | 1/4 | 1/7 | 1/4 | 1 | 1/2 | 1/6 | 1/4 | 1 | 1/2 |
| $V_{1-4}$ | 1/4 | 2 | 4 | 1 | 1/5 | 1/2 | 2 | 1 | 1/5 | 1/3 | 2 | 1 |

**Table 7.** Comparison matrix of relative importance of secondary index risk factors under external factors.

| | $Q_{2\text{-}1}$ | | $Q_{2\text{-}2}$ | | $Q_{2\text{-}3}$ | |
|---|---|---|---|---|---|---|
| | $V_{2\text{-}1}$ | $V_{2\text{-}2}$ | $V_{2\text{-}1}$ | $V_{2\text{-}2}$ | $V_{2\text{-}1}$ | $V_{2\text{-}2}$ |
| $V_{2-1}$ | 1 | 3 | 1 | 2 | 1 | 1/2 |
| $V_{2-2}$ | 1/3 | 1 | 1/2 | 1 | 2 | 1 |

**Table 8.** Comparison matrix of relative importance of risk factors of secondary indicators under the risk of sand mining area.

| | $Q_{3\text{-}1}$ | | | $Q_{3\text{-}2}$ | | | $Q_{3\text{-}3}$ | | |
|---|---|---|---|---|---|---|---|---|---|
| | $V_{3\text{-}1}$ | $V_{3\text{-}2}$ | $V_{3\text{-}3}$ | $V_{3\text{-}1}$ | $V_{3\text{-}2}$ | $V_{3\text{-}3}$ | $V_{3\text{-}1}$ | $V_{3\text{-}2}$ | $V_{3\text{-}3}$ |
| $V_{3-1}$ | 1 | 5 | 3 | 1 | 5 | 2 | 1 | 4 | 3 |
| $V_{3-2}$ | 1/5 | 1 | 1/3 | 1/5 | 1 | 1/3 | 1/4 | 1 | 1/2 |
| $V_{3-3}$ | 1/3 | 3 | 1 | 1/2 | 3 | 1 | 1/3 | 2 | 1 |

**Table 9.** Comparative matrix of relative importance of risk factors of secondary indicators under intrinsic risk of channels.

| | $Q_{4\text{-}1}$ | | | | $Q_{4\text{-}2}$ | | | | $Q_{4\text{-}3}$ | | | |
|---|---|---|---|---|---|---|---|---|---|---|---|---|
| | $V_{4\text{-}1}$ | $V_{4\text{-}2}$ | $V_{4\text{-}3}$ | $V_{4\text{-}4}$ | $V_{4\text{-}1}$ | $V_{4\text{-}2}$ | $V_{4\text{-}3}$ | $V_{4\text{-}4}$ | $V_{4\text{-}1}$ | $V_{4\text{-}2}$ | $V_{4\text{-}3}$ | $V_{4\text{-}4}$ |
| $V_{4-1}$ | 1 | 6 | 3 | 5 | 1 | 7 | 2 | 5 | 1 | 5 | 1 | 4 |
| $V_{4-2}$ | 1/6 | 1 | 1/5 | 1/2 | 1/7 | 1 | 1/4 | 1/3 | 1/5 | 1 | 1/4 | 1/3 |
| $V_{4-3}$ | 1/3 | 5 | 1 | 3 | 1/2 | 4 | 1 | 2 | 1 | 4 | 1 | 2 |
| $V_{4-4}$ | 1/5 | 2 | 1/3 | 1 | 1/5 | 3 | 1/2 | 1 | 1/4 | 3 | 1/2 | 1 |

The same method was used to calculate the weights of risk factors under different indicators in turn, and the weight vectors of the secondary index the stop time of the goaf $V_{1-1}$, the span of the goaf $V_{1-2}$, the area of the goaf $V_{1-3}$ and the depth of the goaf $V_{1-4}$ under the mined-out area risk were obtained, which were (0.554, 0.225, 0.066, 0.155). The second-level indicator of human activity $V_{2-1}$ under external factors and the weight vector of rainfall $V_{2-2}$ was (0.582, 0.418). The second-level indicator of the distance from the slope angle of the sand mining area $V_{3-1}$, the shape of the sand mining pit $V_{3-2}$ and the depth of the sand mining pit $V_{3-3}$ were (0.615, 0.116, 0.269). The weight vector of the secondary index geological structure $V_{4-1}$, groundwater $V_{4-2}$, cohesion $V_{4-3}$ and the slope gradient of side slopes $V_{4-4}$ were (0.509, 0.065, 0.293, 0.132). The consistency test met the requirements, and the final comprehensive weight is shown in Table 10.

**Table 10.** Weights of construction risk factors for poor geological channels.

| First Level Indicator | Weights | Secondary Indicators | Weights | Combined Weights |
|---|---|---|---|---|
| Risk of goaf | 0.555 | The stopping time of the goaf (year) | 0.554 | 0.307 |
| | | The span of the goaf (m) | 0.225 | 0.125 |
| | | The area of the goaf (m$^2$) | 0.066 | 0.037 |
| | | The buried depth of the goaf (m) | 0.155 | 0.086 |
| External factor | 0.055 | Human activities | 0.582 | 0.032 |
| | | Rainfall (mm) | 0.418 | 0.023 |
| Sand mining area risk | 0.233 | The distance from the slope angle of the sand mining area (m) | 0.615 | 0.143 |
| | | The shape of the sand mining pit | 0.116 | 0.027 |
| | | The depth of the sand mining pit (m) | 0.269 | 0.063 |
| Inherent risk of the channel | 0.156 | Geological structure | 0.509 | 0.079 |
| | | Groundwater $\left(\mathrm{L\cdot min^{-1}\cdot(10\,m)^{-1}}\right)$ | 0.065 | 0.010 |
| | | Cohesion of foundation soil (kPa) | 0.293 | 0.046 |
| | | The slope gradient of side slopes (°) | 0.132 | 0.021 |

*4.3. Riskiness Assessment of the Project by Cloud-Based Elements*

After reviewing "*Technical Rules for Design and Construction of Goaf Highways*" (JTG/T D31-03-2011) [42], "*Code for Slope Design of Water Conservancy and Hydropower Projects*" (SL 386-2007) [43], "*The Technical Specification for Supervision and Management of River Sand Mining Planning and Implementation*" (SL 423-2008) [44] and other relevant specifications and a study of the related literature, the hazards of risk-causing factors for the construction of poor geological channels were determined as levels I–V, corresponding to highly dangerous, dangerous, more dangerous, safer and safe, respectively, as shown in Table 11. For quantitative indicators, the specified values were used as the division criteria, and for qualitative indicators, levels I–V were assigned 5, 10, 15, 20 and 25, respectively [38].

**Table 11.** Classification of risk factors for construction risks of poor geological channels.

| First Level Indicator | Secondary Indicators | Hazard Level | | | | |
|---|---|---|---|---|---|---|
| | | I | II | III | IV | V |
| Risk of goaf [13,14,42] | The stopping time of the goaf (year) | (0,1) | (1,2) | (2,3) | (3,4) | (4,10) |
| | The span of the goaf (m) | (160,300) | (120,160) | (80,120) | (40,80) | (0,40) |
| | The area of the goaf (m$^2$) | (2000,3000) | (1600,2000) | (1200,1600) | (800,1200) | (0,800) |
| | The buried depth of the goaf (m) | (0,100) | (100,200) | (200,400) | (400,600) | (600,1000) |
| External factor [42,43] | Human activities | (0,5) | (5,10) | (10,15) | (15,20) | (20,25) |
| | Rainfall (mm) | (300,500) | (150,300) | (100,150) | (50,100) | (0,50) |
| Sand mining area risk | The distance from the slope angle of the sand mining area [43] (m) | (0,3) | (3,5) | (5,7) | (7,9) | (9,15) |
| | The shape of the sand mining pit [44] | (0,5) | (5,10) | (10,15) | (15,20) | (20,25) |
| | The depth of the sand mining pit [45] (m) | (8,10) | (6,8) | (4,6) | (2,4) | (0,2) |
| Inherent risk of the channel | Geological structure [46] | (0,5) | (5,10) | (10,15) | (15,20) | (20,25) |
| | Groundwater [46] $\left(\mathrm{L\cdot min^{-1}\cdot(10\,m)^{-1}}\right)$ | (125,200) | (100,125) | (50,100) | (25,50) | (0,25) |
| | Cohesion of foundation soil [47] (kPa) | (0,20) | (20,30) | (30,40) | (40,50) | (50,100) |
| | The slope gradient of side slopes [48] (°) | (60,90) | (45,60) | (30,45) | (15,30) | (0,15) |

After obtaining the grade division index, we obtained the standard grade cloud model of each risk index according to the method of determining the optimal cloud entropy. The specific numerical characteristic value results are shown in Table 12.

**Table 12.** Numerical characteristics of the cloud model for channel risk level.

| Value \ Risk Level \ Risk Factor Cloud Model | | Highly Dangerous (I) | Dangerous (II) | More Dangerous (III) | Safer (IV) | Safe (V) |
|---|---|---|---|---|---|---|
| $V_{1-1}$ | $E_x$ | 0.500 | 1.500 | 2.500 | 3.500 | 7.000 |
| | $E_n$ | 0.425 | 0.425 | 0.425 | 0.425 | 1.000 |
| | $H_e$ | 0.080 | 0.080 | 0.080 | 0.080 | 0.080 |
| $V_{1-2}$ | $E_x$ | 230.000 | 140.000 | 100.000 | 60.000 | 20.000 |
| | $E_n$ | 59.453 | 16.987 | 16.987 | 16.987 | 6.667 |
| | $H_e$ | 0.080 | 0.080 | 0.080 | 0.080 | 0.080 |
| $V_{1-3}$ | $E_x$ | 2500.000 | 1800.000 | 1400.000 | 1000.000 | 400.000 |
| | $E_n$ | 424.665 | 169.866 | 169.866 | 169.866 | 133.333 |
| | $H_e$ | 0.080 | 0.080 | 0.080 | 0.080 | 0.080 |
| $V_{1-4}$ | $E_x$ | 50.000 | 150.000 | 300.000 | 500.000 | 800.000 |
| | $E_n$ | 42.466 | 42.466 | 84.933 | 84.933 | 66.667 |
| | $H_e$ | 0.080 | 0.080 | 0.080 | 0.080 | 0.080 |
| $V_{2-1}$ | $E_x$ | 2.500 | 7.500 | 12.500 | 17.500 | 22.500 |
| | $E_n$ | 2.123 | 2.123 | 2.123 | 2.123 | 0.833 |
| | $H_e$ | 0.080 | 0.080 | 0.080 | 0.080 | 0.080 |
| $V_{2-2}$ | $E_x$ | 400.000 | 225.000 | 125.000 | 75.000 | 25.000 |
| | $E_n$ | 84.933 | 63.700 | 21.233 | 21.233 | 8.333 |
| | $H_e$ | 0.080 | 0.080 | 0.080 | 0.080 | 0.080 |
| $V_{3-1}$ | $E_x$ | 1.500 | 4.000 | 6.000 | 8.000 | 12.000 |
| | $E_n$ | 1.274 | 0.849 | 0.849 | 0.849 | 1.000 |
| | $H_e$ | 0.080 | 0.080 | 0.080 | 0.080 | 0.080 |
| $V_{3-2}$ | $E_x$ | 2.500 | 7.500 | 12.500 | 17.500 | 22.500 |
| | $E_n$ | 2.123 | 2.123 | 2.123 | 2.123 | 0.833 |
| | $H_e$ | 0.080 | 0.080 | 0.080 | 0.080 | 0.080 |
| $V_{3-3}$ | $E_x$ | 9.000 | 7.000 | 5.000 | 3.000 | 1.000 |
| | $E_n$ | 0.849 | 0.849 | 0.849 | 0.849 | 0.333 |
| | $H_e$ | 0.080 | 0.080 | 0.080 | 0.080 | 0.080 |
| $V_{4-1}$ | $E_x$ | 2.500 | 7.500 | 12.500 | 17.500 | 22.500 |
| | $E_n$ | 2.123 | 2.123 | 2.123 | 2.123 | 0.833 |
| | $H_e$ | 0.080 | 0.080 | 0.080 | 0.080 | 0.080 |
| $V_{4-2}$ | $E_x$ | 162.500 | 112.500 | 75.000 | 37.500 | 12.500 |
| | $E_n$ | 31.850 | 10.617 | 21.233 | 10.617 | 4.167 |
| | $H_e$ | 0.080 | 0.080 | 0.080 | 0.080 | 0.080 |
| $V_{4-3}$ | $E_x$ | 10.000 | 25.000 | 35.000 | 45.000 | 75.000 |
| | $E_n$ | 8.493 | 4.247 | 4.247 | 4.247 | 8.333 |
| | $H_e$ | 0.080 | 0.080 | 0.080 | 0.080 | 0.080 |
| $V_{4-4}$ | $E_x$ | 75.000 | 52.500 | 37.500 | 22.500 | 7.500 |
| | $E_n$ | 12.740 | 6.370 | 6.370 | 6.370 | 2.500 |
| | $H_e$ | 0.080 | 0.080 | 0.080 | 0.080 | 0.080 |

The MATLAB program was prepared according to the three numerical characteristics of the cloud model of the classification level boundaries of each assessment index in Table 12, so that each safety risk level corresponded to a cloud, and five normal clouds were generated using the normal cloud generator. Among them, the standard cloud diagram of the risk factor mining area stopping time is shown in Figure 7, and the standard cloud diagrams of other risk factors were not described in detail.

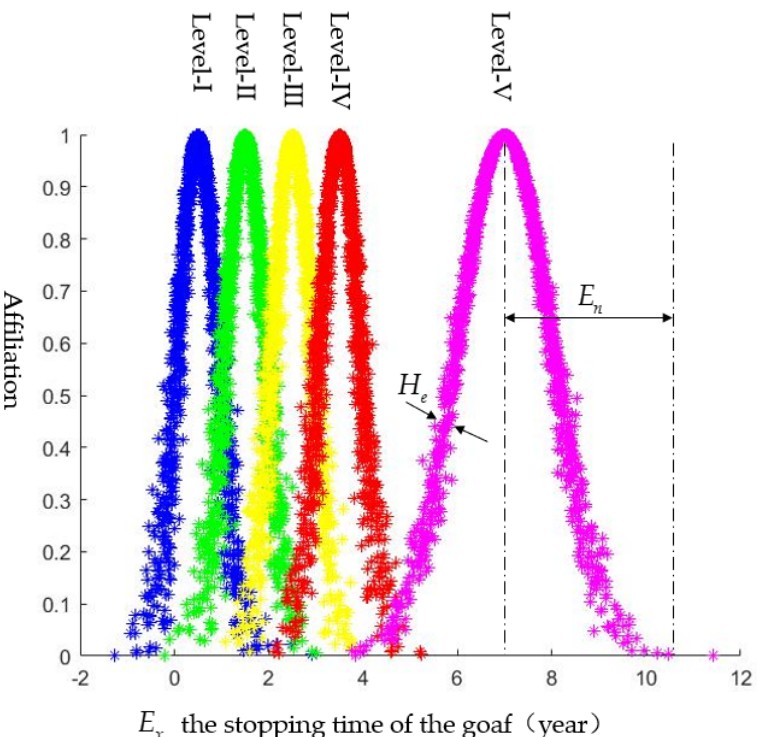

**Figure 7.** Standard cloud diagram of risk factors.

In this paper, three sections of the project were examined in the field, and the measured data were assigned the following values after discussions with professionals.

For each evaluation zone, we input the actual measured values or score values of the 13 evaluation indicators in Table 13, and calculate the cloud correlation degree between each evaluation indicator and the normal cloud of the security risk level standard according to Equation (12). Taking zone $P_1$ as an example, the cloud correlation matrix was calculated as follows.

$$
K = \begin{bmatrix}
0.0001 & 0.0001 & 0.0001 & 0.0001 & 1.0000 \\
0.0918 & 0.0636 & 1.0000 & 0.0647 & 0.0001 \\
0.0184 & 0.0130 & 0.8406 & 0.2100 & 0.0001 \\
0.0635 & 1.0000 & 0.2099 & 0.0002 & 0.0001 \\
0.0001 & 0.0001 & 0.0461 & 0.9730 & 0.0001 \\
0.0008 & 0.0754 & 0.1045 & 0.9728 & 0.0001 \\
0.0001 & 0.0001 & 0.0001 & 0.0001 & 0.5602 \\
0.0001 & 0.0001 & 0.0216 & 1.0000 & 0.0001 \\
0.0001 & 0.0001 & 0.0302 & 0.9715 & 0.0001 \\
0.0221 & 0.0016 & 1.0000 & 0.0021 & 0.0001 \\
0.2137 & 1.0000 & 0.0522 & 0.0001 & 0.0001 \\
0.0002 & 0.0001 & 0.0449 & 0.9939 & 0.1709 \\
0.7772 & 0.8433 & 0.9160 & 0.9995 & 0.1709
\end{bmatrix}
$$

**Table 13.** Measured values of risk factors in different sections of the channel.

| Zone | Measured Values of Risk Factors | | | | | | | | | | | | |
|---|---|---|---|---|---|---|---|---|---|---|---|---|---|
| | $V_{1-1}$ | $V_{1-2}$ | $V_{1-3}$ | $V_{1-4}$ | $V_{2-1}$ | $V_{2-2}$ | $V_{3-1}$ | $V_{3-2}$ | $V_{3-3}$ | $V_{4-1}$ | $V_{4-2}$ | $V_{4-3}$ | $V_{4-4}$ |
| $P_1$ | 7 | 100 | 1300 | 150 | 18 | 80 | 13 | 17 | 3 | 18 | 75 | 25 | 21.8 |
| $P_2$ | 6 | 60 | 900 | 140 | 20 | 80 | 4 | 25 | 3 | 20 | 75 | 25 | 26.6 |
| $P_3$ | 8 | 110 | 1500 | 300 | 20 | 80 | 7 | 15 | 5 | 15 | 75 | 25 | 21.8 |

According to Equation (15), we obtained the sample evaluation result vector $B$ (0.0277,0.1422,0.1973,0.2563,0.3907), and then the weighted average method was used to obtain a single comprehensive evaluation score of 2.172 through Equation (16). In order to reduce the influence of random factors, the calculation was repeated 100 times according to the above steps. According to Equations (17) and (18), the mean and standard deviation of the comprehensive evaluation scores were 2.255 and 0.013, respectively. Finally, the confidence factor was found to be 0.007 according to Formula (19). The factor was small, indicating that the credibility of the evaluation result was high. The calculation methods of the other two zones were the same and the final evaluation results of the three zones are shown in Table 13. At the same time, in order to verify the practicability of the method in this paper, it was compared with the evaluation results of the traditional analytic hierarchy process and matter–element theory model. The specific steps were shown in the literature [49]. This article did not repeat them. The final evaluation results are shown in Table 14.

**Table 14.** Risk assessment results of different evaluation methods for 3 zones.

| Item Number | Method of this Article | | Results of the Evaluation of the Method in the Literature [49] |
| --- | --- | --- | --- |
| | Evaluation Results | Confidence Factors | |
| $P_1$ | IV ($E_{x,r} = 2.255$) | 0.007 | IV |
| $P_2$ | III ($E_{x,r} = 2.730$) | 0.024 | III |
| $P_3$ | III ($E_{x,r} = 2.564$) | 0.019 | IV |

As can be seen from Table 14, the evaluation results of the method used in this paper were basically consistent with those of the literature [49], and the confidence factors obtained by the method in this paper were all smaller, indicating that the cloud model proposed in this paper combined AHP and matter–element evaluation methods for risk assessment with a high confidence level. When comparing the evaluation results, it was found that the final evaluation results of zone $P_3$ were slightly different. According to the expected mean value of zone $P_3$, the evaluation results were relatively close to the scores of grades III and IV, and the obtained standard deviation was small, which indicated that the final results were closer to grade III. Moreover, the weight calculation in this paper combined the decision-making results of multiple experts, and the evaluation model took into account the randomness and ambiguity of the project risk factors. In general, the evaluation results obtained by this method were more reliable. The final risk evaluation level was as follows: the risk level of section $P_1$ was level IV (safer), and the risk level of sections $P_2$ and $P_3$ was level III (more dangerous).

## 5. Conclusions and Outlook

This article integrated the cloud model into the traditional AHP–matter–element theory, to improve the shortcomings of the traditional AHP method and used the aggregation algorithm of the cloud model to bring all the assignments of multi-person decision making into the calculation formula. The determination of the weights of various construction risk indicators was more objective and reliable. At the same time, in order to fully consider the randomness and ambiguity of the channel construction risk classification boundary value, the cloud model was integrated into the matter–element analysis theory to improve the matter–element structure. When building a standard extension cloud model, the original data were directly used, eliminating the normalization process of data and avoiding the possible loss of information. The common use of all three at the same time could not only obtain satisfactory comprehensive evaluation results, but could also provide the credibility information of the evaluation results. Additionally, the model algorithm was simple, adaptable and easy to program and implement. It provided a new method for the comprehensive evaluation of channel construction risks, and provided some help for guiding channel construction.

However, this article used the cloud model to improve the basic analytic hierarchy process when determining the weight. For the actual risk situation of this article, it was more inclined to use the ANP (Analytic Network Process), which can better deal with the network factor system ability. At the same time, it needs to be highlighted that the classification standard of the security risk level of each evaluation index and whether the score value of the evaluation index was reasonable would also have a greater impact on the evaluation result. Therefore, in a subsequent study, it is necessary to further improve the research method, supplement and improve the safety risk assessment index system of multiple influencing factors under adverse geology and study the scoring value of the assessment index and its safety risk classification criteria, in order to further improve the scientific nature of the assessment index system and enhance the comprehensiveness and objectivity of the assessment results.

Finally, for the final evaluation result, the channel project in the article needed to deal with and prevent relevant risk factors before construction. According to the weight, the risk of the goaf and sand mining area should be focused on, in addition to the risk of the goaf. When the mined-out area is stopped, the risk of the sand-mining area needs to be focused on regarding the distance between the sand-mining area and the channel slope; from the one-way factor risk assessment situation, the buried depth of the mined-out area and the local cohesive strength of the soil would be considered. After the corresponding measures were taken, the channel construction would be carried out under the premise of ensuring safety, so as to achieve the goal of sustainable development.

**Author Contributions:** Data collection, Q.L., L.L. and Z.W.; field investigation, Q.L., Z.W. and Q.M.; consulting experts, Z.W.; methodology, Q.L. and Z.W.; data processing, Z.W.; graphic simulation, Z.W. and L.L.; comparison verification Q.L. and Z.W.; conclusion summary, Q.M. All authors have read and agreed to the published version of the manuscript.

**Funding:** This research was funded by the Key Project of Water Conservancy Science and Technology of Henan Province (No. GG202062).

**Institutional Review Board Statement:** Not applicable.

**Informed Consent Statement:** Not applicable.

**Data Availability Statement:** Not applicable.

**Conflicts of Interest:** The authors declare no conflict of interest.

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
