# Peer review of "Construction Risk Evaluation of Poor Geological Channels Based on Cloud Model-Improved AHP–Matter–Element Theory"

_sustainability, doi:10.3390/su13179632_

Round 1

Reviewer 1 Report

In this paper, the authors proposed a Construction risk evaluation of poor geological channels based on cloud model improved AHP-matter-element theory. Mainly the authors have used the AHP matter element theory for risk evaluation. The following comments need to be addressed.

Write down the main contributions of the proposed scheme and the research gap i.e., between the previous works.

Write down the problem formulation i.e., subsection explaining the problem and its formulation.

The authors need to write a table of notations with their explanation.

The introductory paragraphs are very long. Split it into multiple paragraphs with meaningful information.

Write some applications of the AHP in various fields e.g., in software-defined networking it is used for controller selection as given in the following papers.

Quality of Service Improvement with Optimal Software-Defined Networking Controller and Control Plane Clustering (https://www.researchgate.net/profile/Jehad-Ali/publication/348406321_Quality_of_Service_Improvement_with_Optimal_Software-Defined_Networking_Controller_and_Control_Plane_Clustering/links/5ffe605ea6fdccdcb84d789a/Quality-of-Service-Improvement-with-Optimal-Software-Defined-Networking-Controller-and-Control-Plane-Clustering.pdf)

QoS improvement with an optimum controller selection for software-defined networks (https://journals.plos.org/plosone/article?id=10.1371/journal.pone.0217631)

Restrict the tables to one page. Do not split the tables between pages.

Write down the limitations of your approach and future directions in the conclusion section.

Author Response

Response to Reviewer Comments

Dear reviewer,

Thank you very much for your review! We appreciate your suggestions on our research. We wish you all the best! We will answer your questions below. It is important to note that the changes in the text were made through the "Track Changes" feature in Word so that you can see them better, and that the revised line numbers mentioned in the comments below are all under the "Revision" section "No mark'' state corresponding to the number of lines.

Point 1: Write down the main contributions of the proposed scheme and the research gap i.e., between the previous works.

Response 1: After combing through the research of the relevant literature in the preface, the gap and improvement between the newly proposed method and the traditional method are compared in lines 83-94.

After summarizing the applicability of the traditional method in the second part, the shortcomings of the current method are proposed at the end of each paragraph, which are specifically reflected in lines 162-165 and 180-182, and the advantages of the new method are explained in lines 191-204.

Finally, the advantages of the newly proposed method are summarized in the fourth part, which is specifically reflected in lines 629-643.

Point 2: Write down the problem formulation i.e., subsection explaining the problem and its formulation.

Response 2: We re-modified the preface and divided it into 4 parts to elicit and express the problems studied in this article.

At the same time, in the second part, the problems presented in the current research are described in paragraphs, expressing their respective advantages and explaining the current shortcomings, which are specifically reflected in lines 147-165, lines 166-182, lines 183-190, and lines 191-204.

Point 3: The authors need to write a table of notations with their explanation.

Response 3: We added a new table of notations with their explanation in lines 214-215.

Point 4: The introductory paragraphs are very long. Split it into multiple paragraphs with meaningful information

Response 4: We have re-divided the preface into four parts and expressed different content separately, so that it looks clear and meaningful.

Point 5: Write some applications of the AHP in various fields e.g., in software-defined networking it is used for controller selection as given in the following papers.

Quality of Service Improvement with Optimal Software-Defined Networking Controller and Control Plane Clustering (https://www.researchgate.net/profile/Jehad-

Ali/publication/348406321_Quality_of_Service_Improvement_with_Optimal_Software-Defined_Networking_Controller_and_Control_Plane_Clustering/links/5ffe605ea6fdccdcb84d789a/Quality-of-Service-Improvement-with-Optimal-Software-Defined-Networking-Controller-and-Control-Plane-Clustering.pdf)

QoS improvement with an optimum controller selection for software-defined networks (https://journals.plos.org/plosone/article?id=10.1371/journal.pone.0217631)

Response 5: For the traditional AHP, matter-element theory and cloud model, we have separately checked their applications in different fields, which are collectively reflected in lines 156-160, lines 177-180, and lines 187-190, corresponding to references 32-42.

Point 6: Restrict the tables to one page. Do not split the tables between pages

Response 6: We have rearranged the tables in the text to ensure that they are not divided between pages.

Point 7: Write down the limitations of your approach and future directions in the conclusion section

Response 7: We wrote down the limitations and future directions of the research method in this article in the conclusion section. Specifically reflected in lines 644-655.

We sorted out the full text and improved its format and English.

Special thanks to you for your good comments.

Reviewer 2 Report

 In this work, the authors focused on the Construction risk evaluation of poor geological channels.  I believe the data is interesting. However, some issues need to be resolved before accepting this manuscript for publication.

The language of the paper should be stronger regarding grammar and spelling. Some of the sentences are very difficult to understand for the reader. The author should check the language of the paper. There are some errors with use of an/a/the (wrong use or missing). Also, there are some other language issues. For example, in the following lines of the paragraph, there is not a single full stop and at least I can see two different sentences:  

 “Based on the concept  of sustainable development, this paper examines the feasibility of rebuilding channels under adverse geological conditions, and studies whether there are risks and the degree of risk; according to  the characteristics of the expert’s judgment language and the ambiguity and randomness between  various factors, it is proposed The cloud model is used to improve the AHP (Analytic Hierarchy  Process) risk assessment method; at the same time, the traditional matter-element theory is improved through the cloud model, so that the impact of uncertainty and randomness can be comprehensively considered in the evaluation, and finally formed the risk assessment system of cloud based AHP and cloud-based matter-elements.”

There are several other examples as well.

My second concern is on the Preface Section. The preface section needs improvement. This section is too long and still, the motivation of carrying the current research work is not clear from this section. Also, the objectives should be clearly stated in the introduction or preface section. There is a connection problem and it is not clear why the authors conducted this research, what they did and what is the novelty.

Author Response

Response to Reviewer Comments

Dear reviewer,

Thank you very much for your review! We appreciate your suggestions on our research. We wish you all the best! We will answer your questions below. It is important to note that the changes in the text were made through the "Track Changes" feature in Word so that you can see them better, and that the revised line numbers mentioned in the comments below are all under the "Revision" section "No mark'' state corresponding to the number of lines.

Point 1: The language of the paper should be stronger regarding grammar and spelling. Some of the sentences are very difficult to understand for the reader. The author should check the language of the paper. There are some errors with use of an/a/the (wrong use or missing). Also, there are some other language issues. For example, in the following lines of the paragraph, there is not a single full stop and at least I can see two different sentences:  

 “Based on the concept  of sustainable development, this paper examines the feasibility of rebuilding channels under adverse geological conditions, and studies whether there are risks and the degree of risk; according to  the characteristics of the expert’s judgment language and the ambiguity and randomness between  various factors, it is proposed The cloud model is used to improve the AHP (Analytic Hierarchy  Process) risk assessment method; at the same time, the traditional matter-element theory is improved through the cloud model, so that the impact of uncertainty and randomness can be comprehensively considered in the evaluation, and finally formed the risk assessment system of cloud based AHP and cloud-based matter-elements.”

There are several other examples as well

Response 1: We have checked and sorted out the sentences in the full text, corrected some common grammatical errors, and checked and modified the symbols of related sentences, which are specifically reflected in lines 13-21, and some other changes are marked in the text.

Point 2: My second concern is on the Preface Section. The preface section needs improvement. This section is too long and still, the motivation of carrying the current research work is not clear from this section. Also, the objectives should be clearly stated in the introduction or preface section. There is a connection problem and it is not clear why the authors conducted this research, what they did and what is the novelty.

Response 2: In order to link the full text better, we re-divided the preface into four parts, each expressing different content, making it look clear and meaningful. At the same time, the significance of the research is reflected in the beginning, middle and end of the article. After summarizing the applicability of traditional methods in the second part, the shortcomings of the current method are proposed at the end of each paragraph, which are specifically reflected in lines 162-165 and 180-182. The advantages and research significance of the proposed new method are shown in lines 191-204. Finally, the fourth part summarizes the advantages of the newly proposed method, which is embodied in lines 629-643, and also adds the shortcomings of current research and future research directions, which are embodied in lines 644-655 to make the article more complete.

Special thanks to you for your good comments.